# Mechanical Properties of GFRPs Exposed to Tensile, Compression and Tensile–Tensile Cyclic Tests

**DOI:** 10.3390/polym13060898

**Published:** 2021-03-15

**Authors:** Mariana Domnica Stanciu, Horațiu Teodorescu Drăghicescu, Ioan Călin Roșca

**Affiliations:** Department of Mechanical Engineering, Transilvania University of Brașov, B-dul Eroilor 29, 500360 Brașov, Romania; draghicescu.teodorescu@unitbv.ro (H.T.D.); icrosca@unitbv.ro (I.C.R.)

**Keywords:** glass fiber-reinforced-polymer, cyclic loading, tensile, tensile-tensile cyclic test, mechanical properties

## Abstract

Currently there are many applications for the use of composites reinforced with fiberglass mat and fabrics with polyester resin: automotive, aerospace, construction of wind turbines blades, sanitary ware, furniture, etc. The structures made of composites have a complex geometry, can be simultaneously subjected to tensile–compression, shear, bending and torsion. In this paper we analyzed the mechanical properties of a polyester composite material reinforced with glass fiber (denoted GFRP) of which were carried out two types of samples: The former contains four layers of plain fabric (GFRP-RT500) and the second type contains three layers of chopped strand mat (GFRP-MAT450). The samples were subjected to tensile, compression and tensile–tensile cyclic loading. The results highlight the differences between the two types of GFRP in terms of initial elastic modulus, post yield stiffness and viscoelastic behavior under cyclic loading. Thus, it was observed that the value of the modulus of elasticity and the value of ultimate tensile stress are approximately twice higher in the case of GFRP-RT500 than for the composite reinforced with short fibers type GFRP-MAT450. The tensile–tensile cyclic test highlights that the short glass fiber-reinforced composite broke after the first stress cycle, compared to the fabric-reinforced composite in which rupture occurred after 15 stress cycles. The elasticity modulus of GFRP-RT500 decreased by 13% for the applied loading with the speed of 1 mm/min and by 15% for a loading speed of 20 mm/min.

## 1. Introduction

Glass fiber-reinforced-polymer composites (GFRP) are materials used in structural applications [1]. If these types of structures are exposed to cyclic loads, the mechanical properties are degraded as a result of changes in stiffness during cycles. The results of numerous studies have shown that there is a close link between the dimensions and shape of the fibers, their orientation and distribution, and the viscous–elastic behavior of the matrix [2,3,4]. 

Some research papers, such as [2,3,4,5,6] noticed that the stress-strain response of the materials under cyclic loading can be quite different from that under monotonic loading. The nonlinear relationship between stress and strain is governed by the Ramberg–Osgood equation. According to [2], under cyclic loading, the composites behavior recorded the progressive modulus reduction and unsymmetrical straining in tension and compression. In the first cycles of stress, there was a loss of rigidity due to the appearance of perpendicular cracks of the matrix and interfacial debonding initiated by ruptures of glass fibers, air bubbles, gaps and inclusions [3]. Other authors [7,8,9] observed that, under cyclic loading, there is a continuous softening of GFRP, which is usually due to the initiation and increased damage in the matrix, as well as in the fiber ends and in the fiber–matrix interface. There are three stages in the evolution of physical and mechanical damage of GFRP: In the first stage, multiple microscopic cracks develop which lead to a rapid decrease in stiffness; in the second stage, the speed of stiffness loss decreased, but the micro-cracks developed lead to delamination; followed by the last stage, the destruction of the interface between the matrix and the fibers, as well as the rupture of the fibers [9,10,11].

Other studies such as [12,13,14,15,16] have mainly approached the effect of environmental conditions (temperature variation, glass transition temperature, humidity variation, artificial aging) because mechanical deterioration was much more accelerated in the case of the variation of temperature, humidity, UV radiation. Bazli et al. [17] focused on the damaged mechanisms and mechanical performance of FRP bars under elevated temperatures, comparing the GFRP with CFRP, the properties of composites being synthetized from more than 100 experiments. Also, the mechanical cyclic loading causes changes in matrix-fibers linkages or in chemical-physical rearrangements or in van der Waals forces distribution. If these changes exceed the internal cohesion forces or in some areas the chains are weak, the material has viscous-plastic behavior.

Regarding the architecture of glass fibers composites in terms of matrix, fibers sizes, layers arrangement, number of ply, angle between laminas, thickness of composites and the fiber content, the literature reports values of elasticity modulus, stresses and strain in correlation with different parameters [18,19,20]. Nikforooz et al. [21] studied the fatigue performances of samples made from unidirectional E-glass/polyamide 6 pre-pregs composite laminated in comparison with conventional E-glass/epoxy composites, subjecting the studied samples to tensile–tensile fatigue tests that were performed on all laminates until final failure of the specimen. A systematic analysis to clearly determine the effect of creep and recovery on developed damage mechanisms of FRP composites, based on different loading patterns applied to the specimens by changing the cyclic stress levels and stress ratio, as well as interrupting the cyclic loading at different stress levels and different hold times, were presented in [22,23,24,25]. For the determination of the quasi-static tensile strength and fatigue residual tensile strength and probabilistic distributions of a composite laminate, Wu et al. [25] analyzed the behavior of carbon fiber reinforced polymers (CFRP) applying quasi-static tensile and tensile–tensile fatigue tests. The effects of the difference in mechanical properties between stiffness fibers, such as glass fibers and flexible matrices, obtaining an elastomeric composite were investigated in [26,27,28,29]. Regarding the correlation between different fiber orientations of the reinforcement structure and changes of fiber angles related to the load direction, research papers [30,31] pointed out that the minimum elastic modulus was recorded in case of fibers angle orientation of 60°, and the maximum for 0°. Even though the literature is very rich in experimental investigations and notable results, it is observed that little information is available about the structural and mechanical deterioration of composites subjected to tensile–tensile cyclic test, obtaining the hysteresis loops of stress–strain curves. The aim of this research is to analyze the behavior of two types of composite materials reinforced with glass fibers, subjected to tensile, compression and tensile-tensile cyclic stresses in order to obtain the mechanical properties which will be used in simulation of wind turbine blade made from GFRP. The novelty of the paper consists in highlighting the phenomena that takes place in the first cycles of stressing the composite materials reinforced with glass fibers. Thus, the residual deformation generated during the first stress cycle, lead to plastic deformations much higher than those recorded in the subsequent cycles.

## 2. Materials and Methods

### 2.1. Materials

Two types of glass fiber-reinforced-polymer (GFRP) were prepared using a hand lay-up process, for the determination of mechanical properties. The preparation process of the samples consisted in applying the alternatively of resin and fibers (fabric or mat), and with a roller the resin was pressed to enter into the fabrics in order to ensure enhanced interaction between layers [1]. The preparation of samples was provided by the factory S.C. Compozite S.A., Brasov, Romania. The first type of glass fabric reinforced composite (denoted GFRP-RT500) consisted of four layers of plain fabric RT500 with the orientation of the plain weave fabric 0/90/0/90, obtaining five specimens for each test, according to ASTM D 3039/D 3039M and ISO 527-5 (Figure 1a) [31,32]. The second type of GFRP, denoted GFRP-MAT450, consisted of fibers glass chopped strand mat composite, having three alternative layers with the mat density of each layer 450 g/m^2^, 225 g/m^2^ and 450 g/m^2^ (Figure 1b). The reinforcement consisted of discontinuous and randomly oriented fibers glass. For both types of samples, 440- M888 POLYLITE-type polyester resin was used, as matrix, the composites (samples) being obtained by the hand lay-up process. The fiber volume fraction was 50%, in both cases of samples. The physical features of the samples are included in Table 1, and in Table 2, the physical and mechanical properties of the unsaturated polyester resin 440- M888 POLYLITE at 23 °C are presented.

### 2.2. Experimental Setup

#### 2.2.1. Tensile Test

In order to determine the elastic characteristics of the material, the samples were subjected to a static tensile test. In this study, for the analysis of the mechanical behavior of the composites, the specimens were tested on the universal testing machine LS100 Lloyd’s Instrument which is a trademark of AMETEK Test & Calibration Instruments. The equipment belongs to the Mechanical Engineering Department of Transylvania University of Brasov, with a 100 kN load cell. The testing conditions were: environmental temperature (T) 23 ± 3 °C and relative air humidity (RH) 50 ± 5%, according to standard D3039/D3039M [32]. The specimens were loaded with a constant speed of 1 mm/min until breaking. The strains in the tensile direction were measured by means of an extensometer with a gauge length of 70 mm. For data acquisition, Nexygen Plus software was used. The NEXYGEN Plus is the hub of the Lloyd Instruments materials testing systems. After the tensile tests were performed (according to SR EN ISO 527-5 [33]), the characteristic curve, the specific deformation, the longitudinal elastic modulus, and the rupture tensile of each reinforced composite were determined, and on the basis of the load curves, the average deformation energy for each type of sample was calculated. The fracture of samples was analyzed with optical devices.

#### 2.2.2. Compression Test

The tests consisted in the compression of the specimens with loading speeds of 1 mm/min until breaking. The same equipment as in the previous experimental tensile test was used, but without attaching the strain gauges. The compression testing was not performed on the full recommendation of ASTM D6641 [34]. 

#### 2.2.3. Tensile–Tensile Cyclic Test

For the tensile–tensile cyclic test, the samples were subjected to tensile loading in an algorithm test consisting of three series of pulsating cycles (1, 10 and 15 cycles) at 1, 10 and 20 mm/min and maximum load of 2.5 kN and 3.0 kN, respectively. The design test is presented in detail in Table 3. An important aspect in axial cyclic testing is the uniformity of stress and strains in the specimen gage section [6]. Several tests were performed in order to analyze the influence of the loading cycles on the integrity of the composite material in terms of modulus of elasticity, stiffness and storage capacity of the deformation energy. The elastic–viscous–plastic behavior of the composites was also evaluated. This type of test was performed using the same equipment presented above. 

## 3. Results

### 3.1. Mechanical Properties of GFRP under Static Tensile Loading

The differences between GFRP-RT500 and GFRP-MAT450 behavior can be noticed in the superimposed stress–strain curves presented in Figure 2. Comparing the slope of the curves, the GFRP-RT500 tensile curves shows a higher deformation energy storage capacity than the GFRP-MAT450. On the other hand, the higher the slope, the lower is the strain at high values of stress, as it is the case of GFRP-RT500 samples. These type of curves is specific for elastic–plastic materials. From this point of view, the other samples type GFRP-MAT450, revealed a viscous–elastic behavior.

Analyzing the stress–strain curve of the composite reinforced with fabric GFRP-RT500, depicted in Figure 3a, it can be noted that between the elastic limit and the fracture point of the material, there were four areas indicating the interlaminar behavior and the response of the lamina during the loading. Thus, the first area highlighted with the first circle on the curve indicates the first interlaminar micro-crack, the tensile stress being distributed to the rest of the composite (Figure 3a,b). Thus, for the GFRP-RT500 samples, as the applied load increases, more fibers gave way and the load was transferred to the unbroken fibers, which took over a larger amount of the load. The matrix around the unbroken fibers was increasingly loading above the resistance limit, which lead to the yield of the resin layer, as was reported in the literature [35,36,37]. Because of the breaking of the connection between the composite fibers and the matrix, they are not working together and they began to yield successively. This phenomenon leads to the rapid spread of failure and rupture of the composite (Figure 3b). Similar behavior of composite reinforced with plain fabric is reported by [38,39,40]. The micro-cracks caused by the elasticity difference between fibers and matrix are more consistent for the samples GFRP-MAT450 in comparison with GFRP-RT500 (Figure 3c). The samples GFRP-RT500 presented a good correlation between resin tensile elongation and laminate mechanical properties in glass fibers reinforced polyesters. The type of reinforcement in terms of mat density plays an important role in the mechanical behavior. In Figure 3d, it can be seen that the failure occurred near the ultimate tensile strength, characterized by a brittle fracture. The results of uniaxial tensile static tests clearly confirmed that the most resistant GFRP was the one with RT500 reinforcement. In contrast to this, the least resistant was GFRP–MAT450. The mechanical properties of the tested samples, in terms of average values of longitudinal elasticity modulus, stress at break, percentage strain and stiffness are summarized in Table 4. The higher values were recorded by GFRP–RT500, which were almost double compared to GFRP–MAT450. It can be observed that the value of the modulus of elasticity and the value of ultimate tensile stress were approximately twice as high in the case of GFRP-RT500 than for the composite reinforced with short fibers, type GFRP-MAT450. Even if these differences were registered between the two types of composites, in industrial applications both categories of reinforcements are used in different combinations so as to obtain composite structures with high strengths and a homogeneous response in all directions of stress.

### 3.2. Compression Test

In Figure 4 the compressive stress-strain behavior of both GFRP-RT500 and GFRP-MAT450 samples can be noticed, which is very similar. The observed ultimate compressive stress-strains were higher for the GFRP-RT500 samples than for GFRP-MAT450. According to the quasi-homogeneity of composite materials, the behavior of GFRP-RT500 is less convergent and homogeneous as compared to GFRP-MAT450 variation curves. In Table 5 the mechanical properties of GFRP obtained after the compression test are summarized.

### 3.3. Tensile–Tensile Cyclic Test

Figure 5 shows the comparison of hysteresis loops for different loading speed (1 mm/min; 10 mm/min; 20 mm/min). The ultimate tensile strength in case of GFRP-RT500 reached the maximum value for the maximum speed loading (20 mm/min), while GFRP-MAT450 recorded the higher value of ultimate tensile strength for loading speed of 10 mm/min. The changes in the cyclic deformation behavior were more pronounced at the beginning of the cyclic loading (transient behavior) when the plastic deformation reached almost 50% from the total plastic strain at the end of the cyclic loading (Figure 5). Thus, in case of GFRP-RT500, it can be noticed that the plastic strain varied with speed loading: with increases in the speed loading, it decreased the plastic strain and increased the elastic deformation (Figure 5a). The same observation is not applicable for GFRP-MAT450, where the plastic deformation was the same regardless the loading speed (Figure 5b). Wu et al. [39] developed the theory of incremental-secant operator in each phase of elastic–viscous–plastic behavior of composites based on the residual strain and stress. The theory of the schematics of the predictor–corrector incremental-secant formulation in each phase can also be applied in the present study, as can be seen in Figure 5. 

The area within a hysteresis loop depicted in Figure 5 represents the energy dissipated during a cycle; by increasing the number of cycles, the material usually gradually stabilized (steady-state). The yield strength in tensile or compression was reduced after applying a load of the opposite sign that caused inelastic deformation. Appreciable progressive changes were observed in stress–strain behavior during inelastic cycling. It can be considered that GFRP-RT500 is a material made up of two phases: an elastic one represented by glass fibers, and a viscous–elastic one represented by the matrix. The mechanical behavior of these materials could be predicted by the Mori–Tanaka scheme, which was extensively presented in [35,36,37,38,39,40]. The elastic–viscous–plastic behavior of composites as a two-phase materials subjected to tensile – compression was studied experimentally and numerically by Czarnota et al. [40].

In the case of specimens made of GFRP-MAT450, the tests could be done for a single stress cycle. Although the force had a value of approx. 50% of the breaking strength; the composite did not withstand it, and all samples broke in the second cycle. This proved that the GFRP-MAT450 as an own composite was not recommended for use in fatigue applications. Due to the short and dispersed fibers, the deformation energy was not continuously transferred to the mass of the composite, this was taken over and rendered differently by the matrix and by the fibers. In this case, the influence of the elastic viscous and plastic behavior of the matrix was much stronger than in the case of GFRP-RT500 composite. At the monotonic stress–strain, there was no surface damage; by increasing the number of loading cycles with the asymmetry coefficient R = 0, it can be seen the matrix failure and buckling of fibers (Figure 3). Finally, at the 15th cycle, the GFRP samples failed, both matrix and fabric, the fracture region being characterized by cracking, debonding and crushing. At that stage, the material reached the plasticity limit and damage. During cyclic stresses, besides the axial stresses, which were developed in matrix and fibers, at the interfaces between both components, appeared shear stresses due to the different stiffness of each component. So, these stresses lead to the sliding between layers because the elastic limits of matrix and fibers do not overlap. Then, the first fractures from composite appear. The majority of the criteria proposed in the literature identified the following failure modes: Fiber fracture; transverse matrix cracking; shear matrix cracking [8,9,10,11,12,13,14]. The hysteresis curve showed that the decrease of the mechanical properties of GFRP-RT500 occurred along the cyclic loading. The composite lost its strength during the deformation, and the model recorded less stresses to achieve a larger strain in the subsequent cycle by increasing the number of cycles (Figure 6a). Additionally, by increasing the loading speed, the material behavior recorded a decrease of its capacity to accumulate the deformation energy and an increase in the viscous behavior in its nonlinear range (Figure 6).

In Figure 7a–c the values of the main mechanical properties for the studied cases of GFRP-RT500 submitted to cyclic loading are summarized. It can be noticed that the elasticity modulus decreased by 13% for the applied loading with the speed of 1 mm/min, and by 15% for 20 mm/min. The increase of the loading with 20% (from 2.5 kN to 3 kN) leads to the increase of the damage rate of the mechanical properties with almost 6% at the same number of cyclic loadings. The maximum stress was reached for the loading speed of 20 mm/min in both cases of applied forces. The maximum strain of GFRP-RT500 was recorded for the loading speed of 10 mm/min as can be seen in Figure 7c. 

## 4. Conclusions

In conclusion, the paper presents the experimental results of research conducted on two types of glass fiber reinforced polymers (one reinforced with fabric RT500 and the other type reinforced with chopped glass fibers MAT450 and MAT225). The mechanical behavior and the properties of GFRP subjected to tensile, compression and tensile-tensile cyclic test are presented. It can be concluded that:

(1) the GFRP-RT500 samples are characterized by a two times higher longitudinal elasticity modulus (E= 21337 MPa) in comparison with short fibers reinforced composites (GFRP-MAT450, with E=10238 MPa).

(2) Also, the percentage strain of the GFRP-RT500 composite is 1.2 times higher than that of GFRP-MAT450. The rupture strength of the chopped strand mat GFRP-MAT450 is lower with almost 52% in comparison to GFRP-RT500. A similar behavior is noticed for the compression loading. 

(3) The Young’s modulus of plain fabric GFRP-RT500 was 1.82 times higher in comparison with fiber reinforced composites (GFRP-MAT450).

(4) Comparing the mechanical properties of the same composite subjected to different loading (tensile versus compression), it can be noticed that GFRP-RT500 recorded large differences between mechanical properties at tensile test and compression test, in terms of elasticity modulus (7.29 times higher in case of the tensile test) and the rupture strength which was 38% higher for tensile loading than the compression. 

(5) In case of GFRP-MAT500, the differences between tensile and compression mechanical properties were lower (2.83% between rupture strength to the tensile and the compression).

(6) The two types of reinforcement played an important role in elastic–viscous–plastic response at different loading and stresses. Although GFRP-MAT450 proved inferior properties to GFRP RT500, it is known that many of the composite structures also contain short and dispersed fiberglass layers, which have the role in distributing efforts in all directions, ensuring a quasi-isotropy of the composite in plate plan. The presented studies and the obtained values have applicability in the future studies regarding the simulation of the composite structures (wind turbine blades, boats, nacelles, car bodies). 

## Figures and Tables

**Figure 1 polymers-13-00898-f001:**
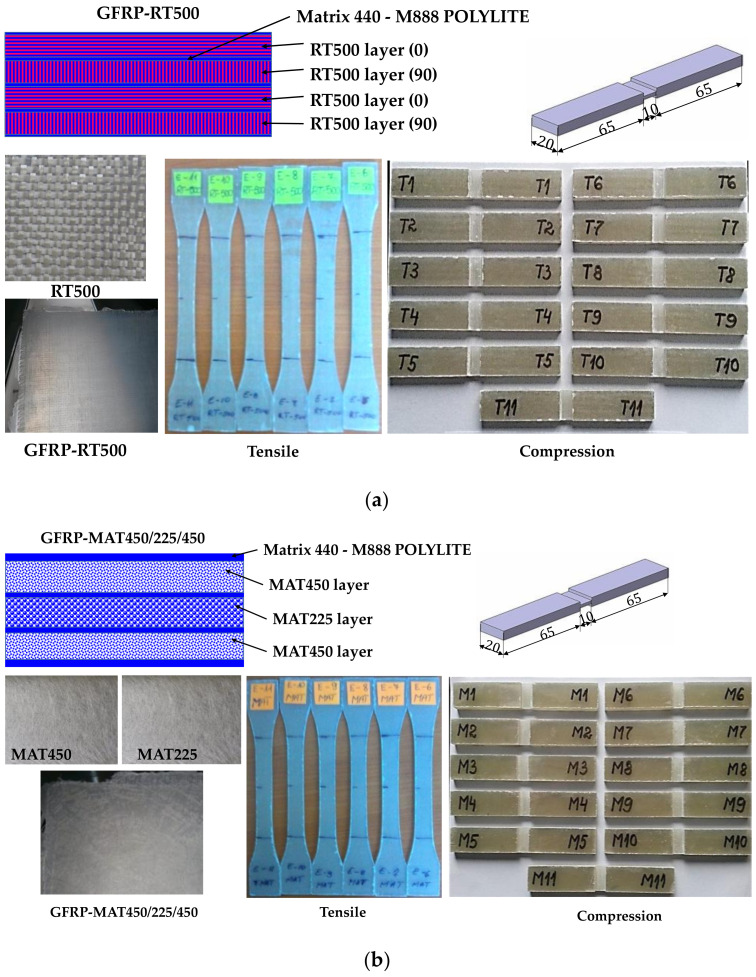
The studied specimens: (**a**) GFRP-RT500 containing four layers of RT500 fabric (0/90/0/90); (**b**) GFRP-MAT450 with three alternative layers.

**Figure 2 polymers-13-00898-f002:**
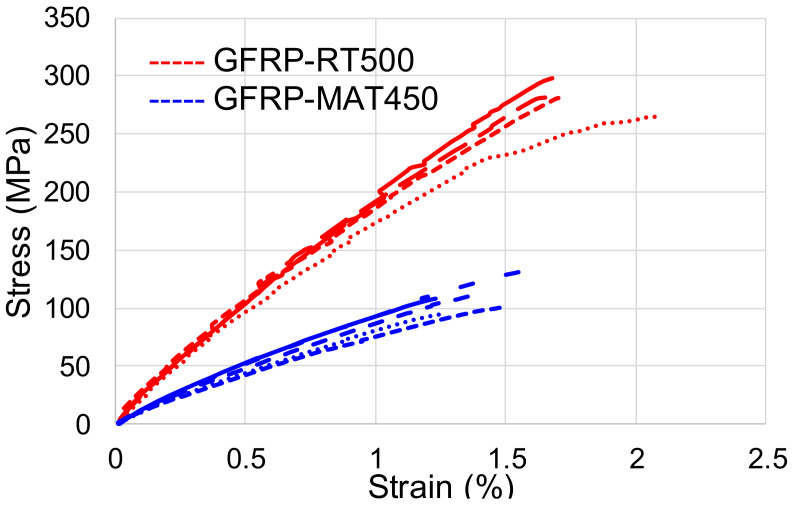
Superimposed stress–strain curves obtained during tensile tests.

**Figure 3 polymers-13-00898-f003:**
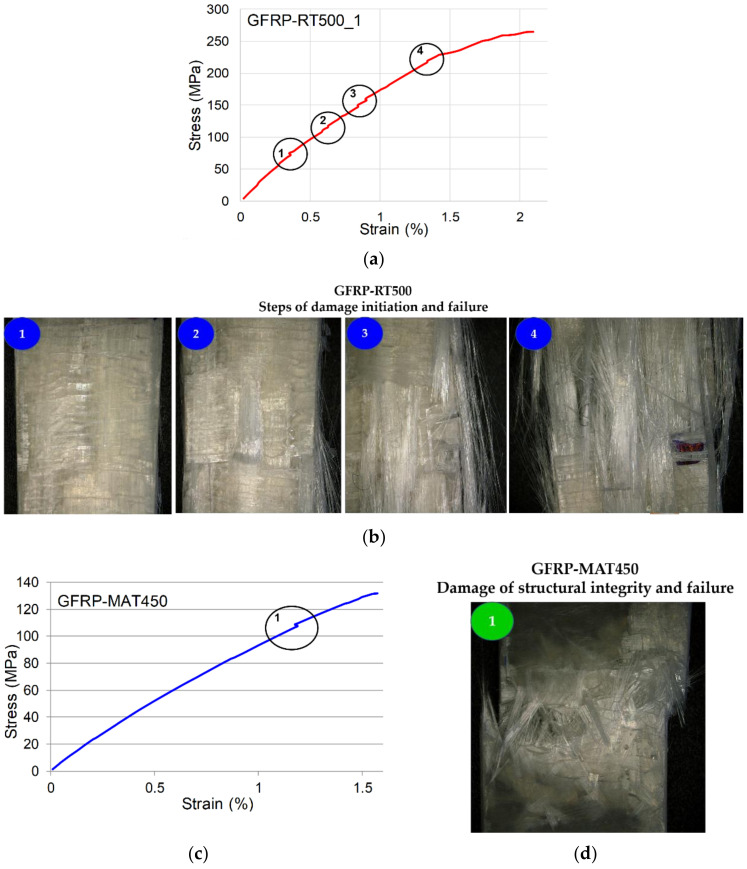
Correlation between stress–strain curves and damage of structural integrity of the two types of GFRP samples: (**a**) GFRP-RT500; (**b**) the microscopic images of damage initiation and failure of GFRP-RT500; (**c**) GFRP-MAT450; (**d**) failure mode of GFRP-MAT450.

**Figure 4 polymers-13-00898-f004:**
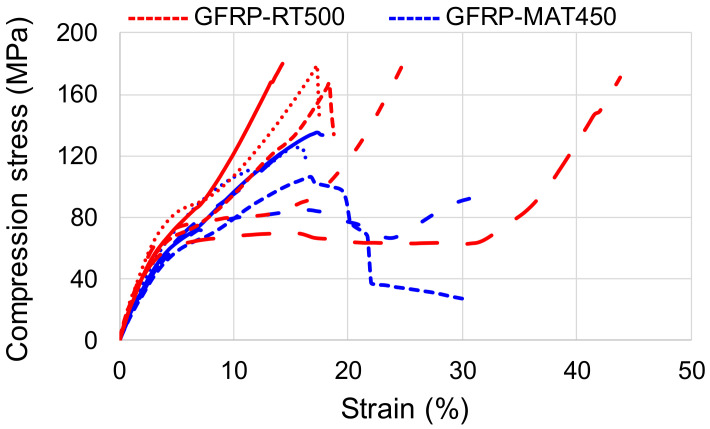
The stress–strain curves in case of compression tests for two types of GFRP samples.

**Figure 5 polymers-13-00898-f005:**
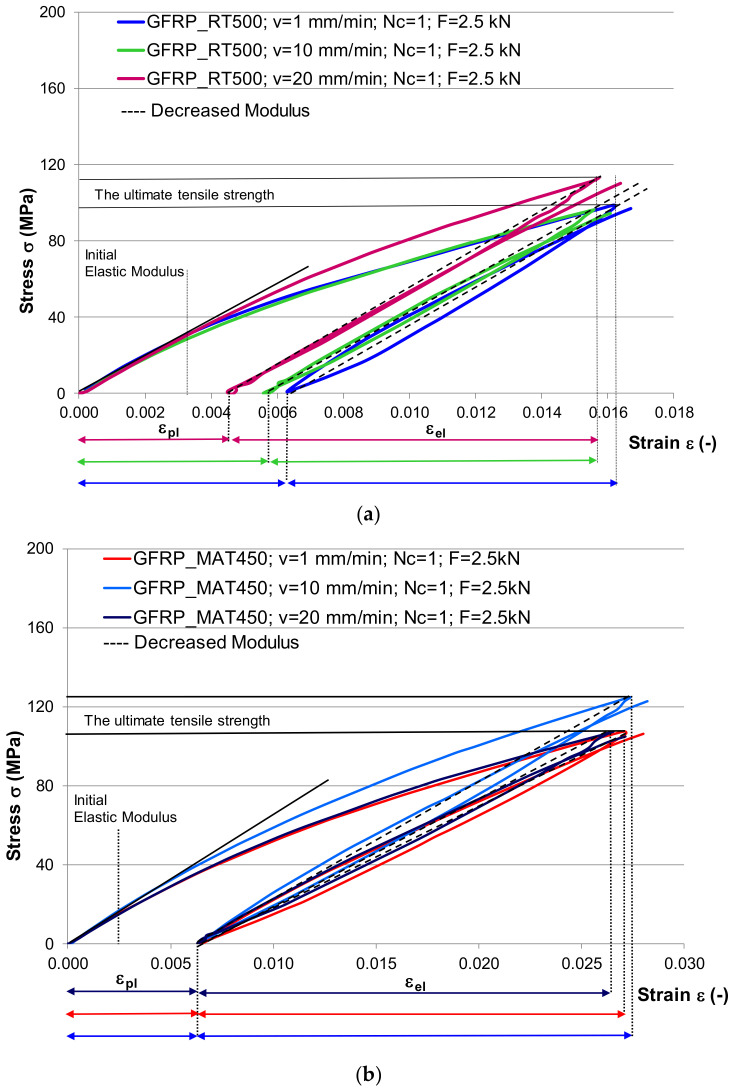
The stiffness loss in fibers direction during first cycle, for different speed loading: (**a**) GFRP-RT500 samples; (**b**) GFRPMAT450 samples (legend: v—speed loading; Nc—number of loading cycles; F—the maximum applied loading; ε_pl_—plastic strain; ε_el_—elastic strain).

**Figure 6 polymers-13-00898-f006:**
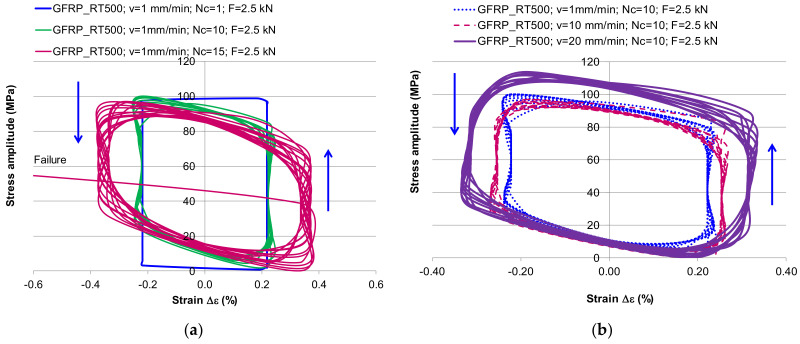
The nonlinear effect of cyclic stresses: (**a**) hysteresis stress–strain curves with increasing the number of cyclic loading; (**b**) hysteresis curves with increasing the loading speed.

**Figure 7 polymers-13-00898-f007:**
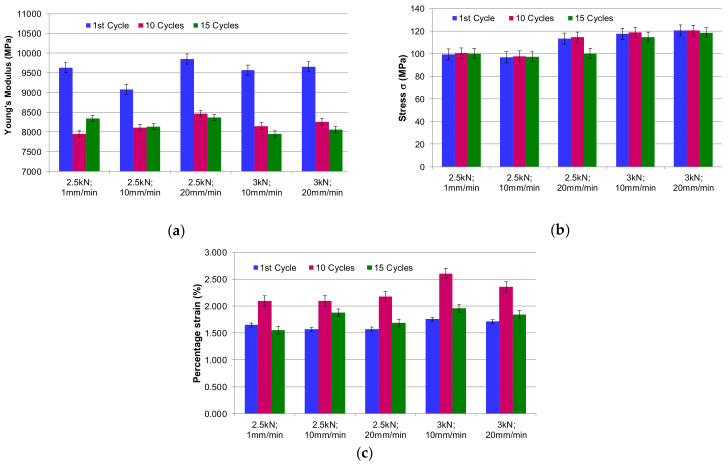
The mechanical properties obtained after tensile-tensile test: (**a**) values of Young’s modulus after different cycles; (**b**) variation of stresses; (**c**) variation of percentage strain.

**Table 1 polymers-13-00898-t001:** The physical characteristics of samples.

Samples	No. of Samples	No. of Layers	Thickness [mm]	Width [mm]	Area [mm^2^]	Gauge Length [mm]
GFRP-RT500 tensile test	5	4	2.50 ± 0.20	10 ± 0.5	25 ± 2.0	70
GFRP-RT500 tensile–tensile cyclic test	5	4	2.50 ± 0.20	10 ± 0.5	25 ± 2.0	70
GFRP-RT500 compression test	5	4	2.50 ± 0.20	20 ± 0.5	48 ± 2.0	10
GFRP-MAT450 tensile test	5	3	2.20 ± 0.20	10 ± 0.5	21 ± 2.0	70
GFRP-MAT450 tensile–tensile cyclic test	5	3	2.20 ± 0.20	10 ± 0.5	21 ± 2.0	70
GFRP-MAT450 compression test	5	3	2.20 ± 0.20	20 ± 0.5	44 ± 2.0	10

**Table 2 polymers-13-00898-t002:** Characteristics of polyester resin type 440-M888 POLYLITE at 23 °C.

Properties	Units	Value	Tested Method
Brookfield Viscosity LVF	mPa·s(cP)	1100–1300	ASTM D 2196-86
Density	g/cm^3^	1.10	ISO 2811-2001
PH (max.)	mgKOH/g	24	ISO 2114-1996
Styrene content	% of weight	43 ± 2	B070
Curing time: 1% NORPOL PEROXIDE 1	Minutes	35–45	G020
Tensile strength	MPa	50	ISO 527-1993
Longitudinal elasticity modulus	MPa	4600	ISO 5271993
Elongation	%	1.6	ISO 527-1993
Bending strength	MPa	90	ISO 178-2001
Elasticity modulus at bending	MPa	4000	ISO 178-2001
The shock resistance P4J	mJ/mm^2^	5.0–6.0	ISO 179-2001
Volumic contraction	%	5.5–6.5	ISO 3521-1976
Glass transition temperature	°C	62	ISO 75-1993

**Table 3 polymers-13-00898-t003:** The tensile–tensile cyclic design tests.

Tensile–Tensile Cyclic Test	Speed (mm/min)	No. of Cycles	Loading (Limit 1) (kN)	Loading (Limit 2) (kN)
Test 1.1	1	1	2.5	0
Test 1.2	1	10	2.5	0
Test 1.3	1	15	2.5	0
Test 2.1	10	1	2.5	0
Test 2.2	10	10	2.5	0
Test 2.3	10	15	2.5	0
Test 3.1	20	1	2.5	0
Test 3.2	20	10	2.5	0
Test 3.3	20	15	2.5	0
Test 4.1	10	1	3.0	0
Test 4.2	10	10	3.0	0
Test 4.3	10	15	3.0	0
Test 5.1	20	1	3.0	0
Test 5.2	20	10	3.0	0
Test 5.3	20	15	3.0	0

**Table 4 polymers-13-00898-t004:** Average values of elastic characteristics obtained after the tensile test. Legend: E—longitudinal elasticity modulus; σ_r_—the rupture strength; ε—percentage strain at maximum load; F_r_—load at break; k—stiffness.

Samples	E [MPa]	STDV E [MPa]	σ_r_ [MPa]	STDV σ_r_ [MPa]	ε [%]	STDV ε [%]	F_r_ (kN)	STDV F_r_ (kN)	k [×10^6^ N/mm]	STDV k [×10^6^ N/mm]
GFRP-RT500	21,337	267	228	53	1.788	0.154	5.738	1.246	7.692	0.246
GFRP-MAT450	10,238	1053	109	15	1.402	0.153	2.5	0.274	3.318	0.109

**Table 5 polymers-13-00898-t005:** Average values of elastic characteristics obtained after the compression test. Legend: E—elasticity modulus; σ_c_—compression strength; ε_r_—percentage strain at break; F_r_—load at break; k—stiffness.

Samples	E [MPa]	STDV E [MPa]	σ_cr_ [MPa]	STDV σ_c_ [MPa]	ε_r_ [%]	STDV ε_r_ [%]	F_r_ [kN]	STDV F_r_ [kN]	k [×10^6^ N/mm]	STDV k [×10^6^ N/mm]
GFRP-RT500	2925	186	165	23	24.15	0.119	7.59	1.255	13.392	0.555
GFRP-MAT450	1556	106	90	50	20.48	0.078	3.93	0.678	7.047	0.231

## Data Availability

The data presented in this study are available on request from the corresponding author.

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
