# Peer review of "Mechanical Properties of GFRPs Exposed to Tensile, Compression and Tensile–Tensile Cyclic Tests"

_polymers, 2021, doi:10.3390/polym13060898_

Round 1

Reviewer 1 Report

This paper describes the tensile, compression and cyclic testing of GFRP consisting of an orientated weave and a chopped strand mat. The information is presented in a logical way and the explanations and conclusions are good.

My main comments are as follows:

  • The English needs correcting in places
  • The naming of the samples is inconsistent throughout the document
  • The novelty of this work should be highlighted. What makes it different from the other papers previously published?

More specific comments are given below:

Abstract

  • Line 17 – It is not clear what these densities refer to.

Introduction

  • The majority of the introduction contains a list of experiments that have previously been published. The authors should consider improving this section by giving the results of these tests rather than describing the test.
  • This list structure to the introduction makes it difficult to read.

Materials and methods

  • Materials – I do not understand what you mean by obtaining 5 specimens in line 94.
  • Section 2.2.3 – 3 cycles and 3 speeds are given however only two maximum loads are listed. It is therefore difficult to see which cycles/speeds these loads refer to.
  • As these are standard tests I feel that figure 2 is unnecessary.

Results

  • Figure 3 - I feel that plots 3a and 3b are not necessary as the same data is shown in figure 3C
  • Line 150 - Please describe how you have determined that RT55 is showing elastic plastic behaviour and MAT450 is showing viscoelastic behaviour.
  • Figure 4 – the images on this plot are unclear
  • Line 187 – what is meant by the break tensile
  • Lines 189-200 – It is mentioned twice in the text that the values for RT500 are almost double that of the MAT and twice that the properties are shown in table 3. Please remove the repeated sections.
  • Table 3 – The standard deviation has only been added for the modulus and tensile strength. Please also include for the other parameters.
  • Table 3 - The values for RT500 don’t seem to match the results shown in figure 3.
  • Figure 7 – Different colours have been used to distinguish between the number of cycles in figure 7a. The same should be done for the speeds in figure 7b.
  • Figure 7 – I am confused how the sample changes between 1 cycle for the blue line and the first cycle in the collection of green lines. If it is the first cycle of the same test then the results should be comparable. Please clarify this.
  • Figure 8 – Figure 8c displays the same set of results as the other two figures however it has been plotted the opposite way around which makes it hard to compare the results. Is there a reason for this?

Author Response

First we would like to thank the reviewers for carefully going through the manuscript and providing helpful suggestions for its improvement. Thanks to their constructive comments, we are able to present clearly and better version than the original manuscript. All the comments of the reviewers have been considered. In particular, the following changes have been made according to the reviewers' suggestions, highlighted by yellow color in the manuscript. 

Reviewer 2 Report

This paper investigate the tensile, compression and tensile-tensile cyclic behavior of two types of glass fibers reinforced composite materials. The main mechanical properties will be used in simulation of wind turbine blade made from GFRP. The paper is well written; however, before publication the following improvements are recommended:

-the authors should better emphasize the novelty of this paper.

-the scale bar must be added to Figure 1.

-the used standards must be added to the reference list.

-more information about the tensile/compression testing machine must be provided: e.g. the maximum value of the load-cell, testing conditions, etc.

-the quality of figure 2 needs to be improved.

-figures 3a and 3b can be deleted, the comparative figure 3c is sufficient.

-for multiplication the “*-asterisk” sign should not be used, but "x-symbol" (see table 3 and others).

-in table 4 there can be no “tensile of rupture” at compression. Only on tensile test does this property exist.

-what is the reason for scattering the results in figure 5?

-the error bars must be added to Figure 8.

-the authors must describe the process of producing the FRP composites. Paper “doi.org/10.3390/polym11101667” presents such aspects in detail. I think it should be taken into account.

-English is not the native language of this Reviewer; however, the manuscript requires some corrections.

Author Response

First we would like to thank the reviewers for carefully going through the manuscript and providing helpful suggestions for its improvement. Thanks to their constructive comments, we are able to present clearly and better version than the original manuscript. All the comments of the reviewers have been considered. In particular, the following changes have been made according to the reviewers' suggestions, highlighted by yellow color in the manuscript. The paper was rewritten taking into account the suggestions. The manuscript was extensively modified
